# Increased Dietary Trp, Thr, and Met Supplementation Improves Performance, Health, and Protein Metabolism of Weaned Piglets under Mixed Management and Poor Housing Conditions

**DOI:** 10.3390/ani14081143

**Published:** 2024-04-09

**Authors:** Joseane Penteado Rosa Gonçalves, Antonio Diego Brandão Melo, Qinnan Yang, Marllon José Karpeggiane de Oliveira, Danilo Alves Marçal, Manoela Trevisan Ortiz, Pedro Righetti Arnaut, Ismael França, Graziela Alves da Cunha Valini, Cleslei Alisson Silva, Nate Korth, Natasha Pavlovikj, Paulo Henrique Reis Furtado Campos, Henrique Gastmann Brand, John Kyaw Htoo, João Carlos Gomes-Neto, Andrew K. Benson, Luciano Hauschild

**Affiliations:** 1Department of Animal Science, School of Agricultural and Veterinarian Sciences, São Paulo State University (UNESP), Campus Jaboticabal, São Paulo 14884-900, Brazil; jp.rosa@unesp.br (J.P.R.G.); diego.brandao@unesp.br (A.D.B.M.); marllon.oliveira@unesp.br (M.J.K.d.O.); danilo.a.marcal@unesp.br (D.A.M.); manoela.ortiz@unesp.br (M.T.O.); pedro.arnaut@unesp.br (P.R.A.); ismael.franca@unesp.br (I.F.); graziela.valini@unesp.br (G.A.d.C.V.); cleslei.silva@unesp.br (C.A.S.); 2Department of Food Science and Technology, University of Nebraska-Lincoln, Lincoln, NE 68588, USA; qinnan@umich.edu (Q.Y.); nkorth2@huskers.unl.edu (N.K.); gomesneto.2@osu.edu (J.C.G.-N.); abenson1@unl.edu (A.K.B.); 3Nebraska Food for Health Center, University of Nebraska-Lincoln, Lincoln, NE 68588, USA; 4Holland Computing Center, University of Nebraska-Lincoln, Lincoln, NE 68588, USA; npavlovikj@unl.edu; 5Department of Animal Science, Universidade Federal de Viçosa, Viçosa 36570-900, Brazil; paulo.campos@ufv.br; 6Evonik Brazil Ltd.a, São Paulo 04711-904, Brazil; henrique.brand@evonik.com; 7Evonik Operations GmbH, 63457 Hanau, Germany; john.htoo@evonik.com; 8Department of Animal Science, Center for Food Animal Health, College of Food, Agricultural, and Environmental Sciences, The Ohio State University, Columbus, OH 43210, USA

**Keywords:** amino acids, haptoglobin, immune challenge, microbiome, multiple origins

## Abstract

**Simple Summary:**

Poor sanitary conditions (SC) in modern swine production can be associated with a variety of factors, including but not limited to: poor biosecurity protocol and management, erratic production logistics/workflow, ineffective cleaning and disinfection protocols, endemic respiratory and enteric diseases, and nutritional program inadequacy for the genotype × environmental interactions present in any given production system. Since nutrition is a large contributor to pig production cost and profitability, and it directly alters pig health individually and at the herd level, it has the potential to alter pig performance under poor sanitary conditions. Threonine, methionine, and tryptophan surplus supplementation did not affect performance nor immune or fecal microbiome parameters on high-sanitary-status pigs housed under good SC. However, our study supports that hypothesis by demonstrating that functional amino acids surplus supplementation improved the performance of piglets housed in a nursery mimicking a poor sanitary condition. Overall, an increased supply of key amino acids can be a viable nutritional strategy for commercial swine production, which is often under immune challenge conditions.

**Abstract:**

A sanitary challenge was carried out to induce suboptimal herd health while investigating the effect of amino acids supplementation on piglet responses. Weaned piglets of high sanitary status (6.33 ± 0.91 kg of BW) were distributed in a 2 × 2 factorial arrangement into two similar facilities with contrasting sanitary conditions and two different diets. Our results suggest that increased Trp, Thr, and Met dietary supplementation could support the immune systems of piglets under a sanitary challenge. In this manner, AA+ supplementation improved the performance and metabolism of piglets under mixed management and poor sanitary conditions. No major temporal microbiome changes were associated with differences in performance regardless of sanitary conditions or diets. Since piglets often become mixed in multiple-site production systems and facility hygiene is also often neglected, this study suggests that increased Trp, Thr, and Met (AA+) dietary supplementation could contribute to mitigating the side effects of these harmful risk factors in modern pig farms.

## 1. Introduction

Poor sanitary conditions (SC) affecting herd health may be associated with improper biosecurity practices adopted in modern swine farms [1]. Poor or suboptimal herd health is often a target of concern due to a diversity of factors that may contribute to a high prevalence of infections, including a variety of pathogen sources and transmission routes [2,3]. In this way, mixing management of pigs from different origins, as occurs in large-scale pig operations using multisite systems, could be a potential source of infectious disease contamination across farms [4]. Since sanitary conditions can largely vary between farms, this practice may trigger the introduction of endemic and epidemic viral and bacterial pathogens, which in turn may lead to illnesses, reducing growth performance and feed efficiency [5,6]. Furthermore, variability in herd health status could be a determinant factor increasing heterogeneity in pig performance within and between modern production farms [7,8]. Considering the improved financial return associated with good sanitary practices [9], mixing management and farm sanitary conditions should be frequent targets of improvement to maintain the herd’s healthy status.

Poor management conditions impair health and growth by generating a cascade of physiological and metabolic changes in pigs [10], including reduced feed intake and immune system activation, which can alter nutrient utilization [11,12]. For instance, immune activation and inflammatory processes can alter feed intake patterns and protein deposition while increasing degradation of body muscle protein to provide amino acids (AAs) for acute phase protein synthesis in the liver [13]. Furthermore, circulating AAs can be repartitioned for the synthesis of immune cells and immune metabolites at the expense of growth performance [14]. The overactivation of the immune system may also increase the release of free radicals, which can overwhelm the steady-state antioxidant capacity [15]. Consequently, the average growth rate of pigs under poor sanitary conditions can be reduced by as much as 30% compared with that of healthy pigs [16,17].

As AAs, such as Thr, Met, and Trp, play essential roles in modulating immune responses and oxidative stress homeostasis [14,18,19,20,21,22,23], their dietary requirements may be increased for weaned piglets raised under poor sanitary conditions. In fact, there is supportive evidence that Thr, Met, and Trp supplementation above dietary requirements may attenuate the negative impact of sanitary challenges on growth performance while promoting the immune status of pigs [11,24,25]. Furthermore, there is the potential for AA to modulate the gut microbiome’s composition, which in turn may be beneficial for host health in terms of nutrient absorption, overall metabolism, and immune functions [26]. Therefore, this study was performed to evaluate the effects of increased dietary Thr, Met, and Trp supplementation on the performance, metabolism, immunological and oxidative status, and fecal microbiota composition of weaned piglets raised under good sanitary conditions or under mixed management and poor housing conditions.

## 2. Materials and Methods

### 2.1. Housing, Animals, and Experimental Design

A total of 144 entire male piglets (Agroceres Pig Improvement Co. [AGPIC Camborough], Rio Claro, SP, Brazil), weaned at 21 days of age, with an initial body weight (BW) of 6.33 ± 0.91 kg, were used in the experiment. The piglets originated from a high-sanitary-status (HSS) multiplier farm. Piglets were allocated in the experiment using a randomized block design in a 2 × 2 factorial arrangement, composed of two SC (GOOD or MM + POOR) and two diets (control (CON), formulated to meet the nutritional requirements according to [27]; or surplus supplementation with Thr:Lys, Met+Cys:Lys, and Trp:Lys ratios adjusted to 20% above the CON diet (AA+)). In total, 12 replicates per treatment with 3 piglets per replicate were used, considering each pen as an experimental unit, except for the fecal microbiome analysis, where each piglet was considered an experimental unit. Figure 1 illustrates the experimental design and shows the distribution of experimental piglets across the two facilities under contrasting SC (GOOD or MM + POOR). The serological results of the health-monitoring program of the high and low sanitary status farms are presented in Appendix A. In summary, the sanitary challenge model featured the mixing of piglets from farms with contrasting sanitary statuses, resulting in housing an additional 48 piglets from a farm with a low sanitary status, following a poor cleaning routine for floor manure removal.

The experiment was carried out during the nursery phase in two similar facilities, each with 36 fully slatted plastic floor pens, for a total period of 42 days. Each pen (2.7 m^2^) had slatted laterals and was equipped with a semiautomatic feeder and a nipple drinker. Both facilities underwent the same cleaning, disinfection, and sanitary interval protocol before piglet housing. As shown in Figure 1, 144 HSS piglets were allocated to one of two facilities under contrasting SC (GOOD or MM+POOR). In this way, 72 HSS piglets (3 piglets per pen; 24 pens in total) were distributed in each facility under GOOD or MM+POOR sanitary conditions. The 12 remaining pens were located between the HSS pens and were kept empty (without piglets) during the experiment in the facility with GOOD SC. In contrast, mimicking mixed management (MM) in the facility with POOR SC, a total of 48 non-experimental entire male piglets (AGPIC 337 × female Camborough), weaned at 21 days of age with an initial BW of 5.43 ± 0.24 kg, obtained from a commercial farm with low sanitary status (LSS), were randomly allocated at 4 piglets/pen into the 12 remaining pens located between pens with HSS piglets. In this way, all HSS pens were paralleled with LSS pens, allowing for physical contact between the piglets through the slatted laterals. In summary, 72 HSS piglets were allocated to the facility with GOOD SC, while 120 piglets were allocated into the facility with MM + POOR SC, of which 72 piglets were HSS and 48 piglets were LSS.

The MM+POOR SC facility had a weekly cleaning routine only to remove the manure located under the slatted plastic floor, which thus represented poor housing conditions. However, in the GOOD SC facility where only HSS animals were allocated, a daily cleaning routine was adopted to remove all manure on and under the slatted plastic floor. Antibiotics were not administered to piglets, regardless of the biosecurity pattern or facility in which they were housed. In addition, each management team was unique to each facility. The teams used clean and disinfected clothing and footwear and protective clothing to enter the facilities in order to avoid cross-contamination among both facilities. A day after ending the experimental period of 42 days, the POOR SC facility was cleaned daily, and experimental and non-experimental pigs were submitted to antibiotic therapy based on Enrofloxacin and Tulathromycin (single dosage). After a week, all pigs were transferred to a commercial farm for the growing and finishing phases.

### 2.2. Experimental Diets

The experimental diets (Table 1) were provided in three feeding phases lasting 14 days each (Phase 1: Days 0 to 14; Phase 2: Days 15 to 28; and Phase 3: Days 29 to 42). All experimental diets were corn- and soybean meal-based. In each phase, HSS piglets were fed the CON diet, formulated to meet the nutritional requirements according to [27], or the AA+ diet, with Thr:Lys, Met+Cys:Lys, and Trp:Lys ratios increased to 20% above the ratios used in the CON diet. LSS piglets were fed the same control diet provided for HSS piglets in each feeding phase during the entire experiment. The diets were formulated using the reported nutrient content, the analyzed AA content of ingredients (NIRS), and the standardized ileal digestibility coefficients (SID) of the AAs, according to [28]. All piglets were given ad libitum access to water and feed throughout the study in both SCs.

### 2.3. Data Collection

#### 2.3.1. Performance

All piglets were weighed individually at the beginning of the experiment and at the end of each feeding phase without fasting. All feed provided and associated leftovers from each pen were weighed weekly. The BW and feed intake of 144 HSS piglets were recorded from Day 0. Thus, BW, average daily gain (ADG), average daily feed intake (ADFI), and gain:feed ratio (G:F) were evaluated for each feeding phase and for the overall experimental period (Days 0 to 42).

#### 2.3.2. Blood Sampling

On Day 0, after fasting for 8 h, blood samples were nonrandomly collected from one piglet per pen (n = 8 for HSS piglets and n = 8 for LSS piglets) between 06:00 and 08:00 h. On Days 21 and 42, after fasting for 8 h, blood samples were collected only from HSS piglets allocated into the GOOD and MM+POOR SC groups. Blood samples were always collected from the piglet with a BW closest to the average BW of the pen on Day 0. The four pens not selected for blood sampling consisted of two pens with the heaviest piglets and two pens with the lightest piglets, totaling n = 8 for HSS piglets for each treatment in both facilities.

A volume of 8 mL of blood was drawn by puncture of the jugular vein into sterile tubes containing EDTA (BD Vacutainer, São Paulo, SP, Brazil). On Day 0, shortly after blood drawing, a fraction of the blood aliquot was used for total and differential leukocyte counting (eosinophils, neutrophils, lymphocytes, and monocytes) by an automated Microcell counter (Horiba Micros-60; Horiba ABX SAS, Montpellier, France). Blood samples collected on Days 0, 21, and 42 were kept refrigerated on ice and centrifuged at 1500× *g* for 10 min at 4 °C (Novatecnica^®^, NT 835, Piracicaba, SP, Brazil) to obtain serum and plasma. Subsequently, both serum and plasma aliquots were stored at −80 °C for subsequent analyses of the serum concentration of acute phase proteins, metabolites, and plasma oxidative parameters.

#### 2.3.3. Analysis of Acute Phase Proteins and Biochemical and Redox Parameters

Serum acute phase proteins (immunoglobulin A (IgA), immunoglobulin G (IgG), ceruloplasmin, transferrin, albumin, haptoglobin, and α1 acid glycoprotein) were analyzed by electrophoresis in sodium dodecyl sulfate–polyacrylamide gels [29], and the concentrations were normalized by the total protein serum concentration (Labmax Plenno; Labtest Diagnostics SA, Lagoa Santa, Brazil). Serum concentrations of total protein, urea, creatinine, lactate, and lactate dehydrogenase (LDH) were analyzed in a high-performance semiautomatic spectrophotometer using the biuret method for biochemical tests (Labmax Plenno; Labtest Diagnostics SA, Lagoa Santa, Brazil), with the support of Labtest commercial reagents^®^. Blood plasma on Day 21 was used to evaluate the piglets’ oxidative status according to the following parameters: reactive oxygen species (ROS), lipid peroxidation (LPO), total antioxidant capacity (T-AOC), superoxide dismutase (SOD), glutathione S-transferase (GST), reduced glutathione (GSH), glutathione disulfide (GSSH), and GSH:GSSH ratio. T-AOC was evaluated by the ferric-reducing ability of plasma (FRAP) according to the method of [30]. SOD activity was determined by a spectrophotometric method [31]. The ferrous oxidation in xylenol orange (FOX) assay was used for the detection of plasma LPO [32]. A fluorometric method was used to determine plasma GSH and GSSH concentrations [33]. The Nernst equation (Eh = −264 − (61.5/2) × log GSH2/GSSG) was applied to determine the redox potential of the GSH:GSSG ratio. In addition, a fluorometric method was also used for the analysis of plasma ROS determination [34]. GST was determined with the conjugation of GSH with 1-chloro-2,4-dinitrobenzene catalyzed by GST [35].

#### 2.3.4. Fecal Sampling and DNA Extraction

Fecal samples were always collected from the HSS and LSS piglet with the BW closest to the average BW of the pen on Day 0 (n = 11). All fecal DNA extractions were conducted using the Qiagen DNeasy^®^ PowerSoil^®^ Pro kit according to the manufacturer’s guidelines. All samples were eluted in 60 µL of elution buffer and frozen at −80 °C prior to quality assessment and sequencing. All samples were quality checked for DNA concentration using a Nanodrop One spectrophotometer (Thermo Fisher Scientific c Inc., Middletown, VA, USA).

#### 2.3.5. 16S rRNA Amplicon Sequencing and Bioinformatic Processing

The V4 region of the bacterial 16S rRNA gene was amplified from each sample using Phusion High-Fidelity PCR Master Mix with HF Buffer (Thermo Scientific, Waltham, MA, USA), following the modified dual-indexing sequencing strategy [36,37]. Paired-end sequences were analyzed using the Quantitative Insights Into Microbial Ecology (QIIME) program (version 2, 2021.2) [38]. Sequences were truncated (220 bases for forward reads and 160 bases for reverse reads) and denoised into amplicon sequence variants (ASVs) using DADA2 [39], and then rarefied to 5000 reads per sample. A total of 3,384,304 sequences were reads, with an average of 25,638 reads per sample for the bacterial 16S rRNA analysis (Appendix A). All ASVs were assigned taxonomic information using the prefitted sklearn-based taxonomy classifier SILVA database (release 138) [40,41]. All sequences can be found on NCBI (https://www.ncbi.nlm.nih.gov/bioproject/PRJNA985664, accessed on 20 June 2023). Prior to statistical analysis, only bacterial taxa (those containing the domain “Bacteria”) containing genus-level information in the name (g from QIIME2 output) were filtered for both diversity and taxonomic analyses.

### 2.4. Network Analysis of the Fecal Microbiome Data

Network construction and visualization were performed using the NetCoMi package [42] within the R (v4.2.2) statistical framework. Overall, the samples were normalized to 5000 reads and grouped by sampling time point, SC, and diet. Each network was constructed for each group or identified treatment by employing a centered log-ratio transformation and Spearman-based correlations between the 50 most abundant microbial taxa. Data interpretation was summarized based on each node representing a bacterial taxon, and the size of each node was scaled according to its centrality. Edges represent associations between taxa, with positive associations colored green and negative associations colored red; the thickness of each edge corresponds to the strength of the association. Edges representing a value less than 0.5 are not shown. Taxa were colored based on clusters calculated in the network construction. The three taxa with the highest degree of centrality were selected to highlight cluster-specific central taxa. When the sample size was limited, the data were excluded from the analysis.

### 2.5. Statistical Analyses

For pig performance and metabolic, immunological, and oxidative parameters, data normality was verified using the Shapiro–Wilk test, and studentized residues were used to verify outliers. Subsequently, data were analyzed using the GLIMMIX procedure of SAS [43] and presented as least squares and means. The mixed model used for data analysis from HSS piglets following the factorial design included the fixed effects of SC, diets (Ds), and their interactions (SC × D). The effect of the diet was also analyzed separately within each SC. For mixed models, the blocks of BW were included as a random effect. The results of the analysis of acute phase proteins and biochemical parameters on Day 0 were used as covariates for Days 21 and 42. The model used for data analysis of leucocyte counts included a fixed effect for sanitary status (HSS vs. LSS piglets) on Day 0. No other responsive parameter was evaluated for LSS piglets during the experiment, except for comparing HSS and LSS fecal microbiome alpha- and beta-diversity (D21 and 42). LSS piglets are non-experimental animals that were allocated into the poor SC facility only to compose the mixed management as part of the challenge model. The effects were considered significant, with values of *p* ≤ 0.05 and trends with values of *p* ≤ 0.10.

All bacterial taxonomic outputs from quality control to statistical modeling were processed using R version 4.0.5. Only bacterial genus-level annotation was used in all statistical analyses. In brief, the tidyverse library (version 1.3.1) was used for data exploration, analysis, and plotting. All analyses were conducted by separating the two facilities differing in SC: “GOOD” vs. “POOR”. For alpha-diversity (Shannon’s and Simpson’s D indices), Welch’s two-sample T test (two treatments only—diet or SC effect) was used to assess significance (*p* ≤ 0.05). Both the Shannon’s and Simpson’s D indices of alpha-diversity were calculated with the diversity function in R from the vegan library (version 2.6.2). The Welch two-sample t test function used was from the stats library (version 4.0.5).

Beta-diversity was calculated using the vegdist function from the vegan library in R (version 2.6.2) while using a Bray–Curtis distance matrix and removing missing values of the analysis. For the principal component analysis (PCA), a classical multidimensional scaling model was used to reduce the data to two dimensions (2 principal components—PCs) using the cmdscale function (k = 2) in R from the stats library (version 4.0.5). PERMANOVA was used to calculate the effect of treatment on beta-diversity (Bray–Curtis distances) using the adonis2 function in R from the vegan library (version 2.6.2).

Additionally, for two-group differences in taxon relative abundances, a Welch two-sample T test function was used. However, both the false discovery rate (FDR) and Bonferroni correction (BC) were applied to reduce false discoveries. Only the taxa that passed the significance level for both the FDR and BC at the end points (Day 21 and Day 42) were considered significantly different between diet groups for the HSS animals (*p* ≤ 0.05). Ultimately, differences between the mean relative abundance for each family/genus/species (taxon) across groups were depicted as a heatmap using the log_2_-transformed relative abundance of the most dominant taxa based on a 2% cutoff and network analysis of clusters of co-occurrences of the most relevant taxa. For taxa with zero mean counts for a given treatment, the value of 1 was added prior to log_2_ transformation being applied. For all R functions used, when not stated, the default parameters were used for calculations. Due to the individuality of the gastrointestinal microbiome composition in animals, each piglet was considered an experimental unit for the analysis [44].

## 3. Results

### 3.1. Characterization of the Health and GI Microbiome Status

On Day 0 (at arrival), piglets from the LSS farm had, on average, higher counts of total leukocytes (*p* < 0.01), granulocytes (*p* < 0.01), and monocytes (*p* < 0.01) than those in piglets from the HSS farm (Table 2). However, there was no significant difference in the lymphocyte count between HSS and LSS piglets (*p* > 0.05). GI microbiome analysis further revealed no significant differences in alpha-diversity (Shannon’s and Simpson’s D indices) between HSS and LSS piglets on Day 0 (Figure 2A,B, respectively). Notably, LSS piglets appeared to have started the trial with lower alpha-diversity, although the *p* value was not significant (Figure 2A,B). However, beta-diversity analysis captured different clustering on Day 0, separating HSS vs. LSS piglets (*p* < 0.005, R-squared = 0.12) (Figure 2C). Multiple taxa showed significant differences between HSS and LSS on Day 0, but only *Oribacterium* passed the family-wise *p* value correction thresholds after accounting for the number of comparisons made between groups (*p* = 0.00014, FDR = 0.0333, Bonferroni = 0.0333). A posteriori (Days 21 and 42), no taxon was found to be significantly different between HSS and LSS piglets allocated to the MM + POOR SC. In addition, the serological results of the health-monitoring program conducted for high- and low-sanitary-status farms are presented in Appendix A.

### 3.2. Growth Performance

There was no interaction (*p* > 0.05) between SC and D on growth performance variables in any of the experimental phases evaluated (Table 3). However, HSS piglets under GOOD SC had higher final BW, ADFI, and ADG than HSS piglets under MM + POOR SC in all experimental phases (*p* < 0.05). Furthermore, throughout the experiment (Days 0 to 42), piglets under GOOD SC had higher final BW, ADFI, ADG, and G:F than piglets under MM + POOR SC did (*p* < 0.05). Within each SC, piglets under MM + POOR SC fed AA+ showed a trend (*p* < 0.10) of higher final BW, ADFI, and ADG compared to those of piglets fed the CON diet during Phase 2. During Phase 3, and for the entire nursery phase, piglets under MM + POOR SC fed the AA+ diet had greater (*p* < 0.05) final BW, ADFI, and ADG compared to those of CON diet piglets.

### 3.3. Acute-Phase Proteins, Biochemical and Redox Parameters

There was no interaction (*p* > 0.05) between SC and D for any of the acute phase proteins analyzed on Days 21 and 42 (Table 4). On Day 21, the average serum concentration of haptoglobin, a positive acute phase protein, in piglets under MM + POOR SC was approximately 44% higher (0.35 vs. 0.80 mg/mL, *p* < 0.05) than that in piglets under GOOD SC. On Day 42, haptoglobin concentrations were similar between SC (*p* > 0.05), whereas the average serum concentration of albumin, a negative acute-phase protein, in piglets under MM + POOR SC was lower (33.8 vs. 31.2 g/L, *p* < 0.05) than that in piglets under GOOD SC.

There was no interaction (*p* > 0.05) between SC and D and non-SC effects for any of the biochemical variables analyzed on Days 21 and 42 (Table 5). However, on Day 21, piglets under MM + POOR SC fed the AA+ diet had lower (*p* < 0.05) serum urea and creatinine concentrations than those in CON diet piglets (16.35 vs. 27.13 mg/dL and 1.04 vs. 1.19 mg/dL, respectively). On Day 42, piglets under MM + POOR SC fed the AA+ diet had lower (*p* < 0.05) serum urea and lactate concentrations than those in CON diet piglets (19.21 vs. 26.06 mg/dL and 85.75 vs. 48.71 mg/dL, respectively). On this same day, AA supplementation tended to increase LDH serum concentration in GOOD SC piglets (*p* = 0.09), with concentrations, on average, being 1256 vs. 1403 U/L in piglets fed the AA+ diet compared to the CON diet, respectively.

There was no significant interaction (*p* > 0.05) between SC and D for any of the oxidative metrics analyzed on Day 21 (Table 6). However, the blood plasma of piglets under MM + POOR SC showed a tendency (*p* = 0.08) toward increased circulating ROS levels. Moreover, SOD activity was higher (*p* < 0.05) in pigs under GOOD SC than in pigs under MM + POOR SC. Regarding the diet effect, pigs under GOOD SC fed the AA+ diet had lower GSH:GSSG Eh values (a lower value, i.e., closer to zero, indicates a less oxidized redox status) than that in piglets fed the CON diet (−316.9 vs. −324.3, *p* = 0.05).

### 3.4. Fecal Microbiome Compositional Characteristics

Fecal microbiome analysis was conducted only on a subset of animals that were not randomly selected after the conclusion of the experiment. Overall, fecal microbiome analyses revealed no major significant differences (*p* > 0.05) in bacterial community composition or individual taxon abundances at the bacterial genus level across treatments over time. The alpha-diversity analysis was performed using both Shannon’s and Simpson’s D indices as standards. Figure 3A,B and Appendix A show the absence of significant differences between diets for GOOD and MM+POOR SC (*p* > 0.05) when measured by Shannon’s or Simpson’s D indices of alpha-diversity (*p* > 0.05), respectively. Furthermore, no significant changes were found in community dispersion or clustering when using a beta-diversity analysis according to diet in GOOD and MM + POOR SC (Figure 3C) (*p* > 0.05). Additional community structure analysis based on co-occurrence network mapping revealed the absence of major temporal changes in architecture or keystone nodes (taxa) for HSS animals housed either in GOOD or MM + POOR SC, despite surplus AA+ supplementation (Figure 4, Figure 5 and Figure 6). Further statistical analysis was carried out to assess taxonomic compositional differences across dietary groups for HSS animals only in both the GOOD and MM + POOR SC while accounting for multiple comparisons (FDR and BC statistical corrections), which ultimately revealed no specific taxon or taxa that were preferentially enriched or diminished over time (*p* > 0.05). Therefore, Figure 7 depicts the major core taxa present across treatments over time while capturing the overall stability in microbial composition, as measured using 16S rRNA sequencing analysis.

## 4. Discussion

Poor sanitary conditions may be found in commercial swine operations that have inadequate biosecurity protocols, facilities, husbandries, nutritional programs, and endemic respiratory and enteric diseases [1,3]. Nutrition accounts for up to 75% of swine production costs [45] and has the potential to impact individual and herd-based health by maximizing growth and productivity, enhancing host defenses, and altering the GI microbiome [46]. For instance, dietary supplementation with Thr, Met, and Trp has been studied and deployed to enhance the growth and feed efficiency of pigs under sanitary challenges [24,25]. Increasing dietary supplementation of Thr, Met, and Trp in 20% related to Lys was carried out following Valini et al. [24] and Rodrigues et al. [25], in an attempt to achieve similar beneficial results for performance and health as those found by these studies; however, we used piglets under mixed management and poor sanitary conditions. Thus, the present study evaluated the effects of surplus dietary Thr, Met, and Trp supplementation and sanitary conditions on performance, physiological, metabolic, immunological, and oxidative metrics, as well as GI microbiome characteristics and the composition of the piglets. Overall, our study demonstrated that Thr, Met, and Trp supplementation can ameliorate the performance of high-sanitary-status weaned piglets under poor sanitary conditions (mixed piglets and poor housing conditions) while having no effect on counterpart piglets raised under GOOD SC. HSS and LSS farms were specifically selected based on the sanitary protocols adopted in each farm. A multiplier farm that followed a rigorous sanitary protocol was selected for the HSS origin, while a commercial farm with poor sanitary status resulting in poor performance parameters was selected for the LSS origin. When establishing these conditions to characterize the challenge model, a moderate and recurrent inflammatory process was expected in LSS piglets, which was confirmed by initial immunological screening and fecal microbiome compositional assessment on Day 0. LSS piglets were housed only to compose the challenge model, and additional evaluation (D 21 or D 42) for performance or health was not in our interest. Since weaned piglets from different origins may be mixed under poor housing conditions in multiple-site large-scale swine operations, this study points toward Trp, Thr, and Met having the potential to be strategically deployed to enhance pig performance for piglets undergoing sanitary challenges.

The absence of effects of the AA+ diet on the growth performance of piglets raised under GOOD SC indicates that the nutrient requirements were met in the experimental diets. However, an eventual positive response to the AA+ diet in GOOD SC may be associated with the effect of Trp stimulating the appetite through enhancement of ghrelin expression and secretion [47]. In addition, Trp, Thr, and Met play functional roles in supporting pig health and are especially important for the synthesis of defense molecules related to aiding the integrity of the gut mucosal barrier and regulating the antioxidant and immune responses of challenged pigs [7,48]. In view of this, the model used for sanitary challenge in this study resulted in reduced performance and metabolic changes that may affect the Thr [20], Met [22], and Trp [23] requirements. Reduced plasma Trp availability during the inflammatory process has been attributed to the synthesis of acute phase proteins such as haptoglobin, which have a high Trp content in their composition, and to the increased catabolism of Trp to kynurenine [12,49]. Kynurenine has antimicrobial properties that restrict the proliferation of bacteria, viruses, and parasites [50]. Furthermore, the effects of Trp on piglet performance have mainly been due to its influence on feed intake [51]. For instance, immune response stimulation and inflammatory processes affect the redox status, which demands greater amounts of sulfur AAs (Met + Cys) to reestablish antioxidative homeostasis [52]. In addition, Thr is an indispensable AA for mucin synthesis, which is crucial for preventing pathogen adhesion on gut mucosa, to maintain gut health [19]. Based on these aforementioned functions, the growth performance of MM + POOR SC piglets was attenuated by the AA+ diet, suggesting a beneficial role of Trp, Thr, and Met surplus supplementation in supporting health [11,25,53].

Challenge models creating poor sanitary conditions for pigs, as used here and by others [11,24,54,55,56,57,58], could face strong difficulties in standardization. For instance, differences in the resistance/resilience due to genetic- and sex-related factors, the influence of the sanitary conditions of farms considered to be “mixed” on piglet health status, the pathogenic potential of stool used to dirty the facility, the facility sanitation level before beginning the experiment, the variations in pathogenic threshold pressure, and bacterial strain pathogenicity are factors affecting the standardization of models. However, the present study has presented the negative impact of poor sanitary conditions on pig health and growth while presenting the potential use of Thr, Met, and Trp surplus supplementation in supporting health and attenuating growth reduction, even at contrasting intensities of challenge. Together, the literature data have reinforced that Thr, Met, and Trp are essential for improving the physiological and metabolic responses of challenged pigs.

Due to mixed management and poor sanitary conditions affecting performance during the weaning period, some serum and plasma biomarkers were measured to quantify the dietary effect on the piglets’ metabolic changes and immune and redox responses. Notably, an increased serum concentration of haptoglobin is a relevant indicator of ongoing inflammatory processes [11] by pathogenic infections [59]. Moreover, increases in serum haptoglobin levels have been correlated with higher variability in ADG, likely due to the redistribution of nutrients used for muscle deposition being fated to immune molecule biosynthesis [10]. Albumin is also a key health status biomarker, whereby its serum concentration tends to decrease in piglets facing an inflammatory process [25,57], while oxidative metabolism and free radical formation (reactive oxygen species—ROS) can be imbalanced under such circumstances [60,61]. Free radicals are molecules that contain oxygen atoms with an unpaired electron and have strong bactericidal properties [62]. On the other hand, oxidative stress causes the oxidation of biomolecules such as lipids, proteins, and DNA, as well as the loss of biological functions and/or homeostatic imbalance [63]. For instance, the higher haptoglobin levels, lower albumin synthesis, and the tendency for higher circulating ROS in piglets under MM + POOR SC are indications of changes in immune and redox parameters. As such, the AA+ diet tended to reduce haptoglobin levels in piglets under MM + POOR SC, which may in part explain the supplementation effect in attenuating POOR SC pigs’ reduced growth performance.

Additionally, metabolic changes were marked by the AA+ diet’s effect on urea and creatinine serum concentrations in the MM + POOR SC group. Urea is a primary metabolite derived from the catabolism of AAs, while creatinine is an indicator of protein degradation in skeletal muscle [64]. Both metabolites have been associated with reduced efficiency of AA utilization and lower protein synthesis during immune system activation [47]. Consequently, AA supplementation can attenuate the immune impact on muscle protein mobilization and degradation in pigs [64]. In fact, piglets under MM + POOR SC fed AA+ had lower serum concentrations of urea (on Days 21 and 42) and creatinine (on Day 21) than those in CON diet piglets. While it was previously considered a metabolic waste product, the metabolic role of lactate has been reconsidered to better understand its involvement as a pro- or anti-inflammatory marker mediating immune cell metabolism [65]. Amino acids that are released from muscle protein breakdown or from the diet may be used as an energy source to support immunometabolism. The inflamed tissue milieu may also change the energy metabolism, increasing lactate production by aerobic glycolysis and attempting to regulate cell immune responses. Accordingly, piglets under MM + POOR SC fed AA+ had a lower lactate concentration than that in CON diet piglets. This finding suggests that an AA+ diet may reduce the changes in cell metabolism, attenuating the impact of sanitary challenge on the immune response, likely by increasing feed intake and, thus, blood AA availability, providing a better AA balance for immunometabolism.

The adoption of good husbandry and sanitary practices during weaning transition is essential for avoiding severe GI dysbiosis, which can lead to diarrhea and reductions in feed efficiency during the nursery stages of production [66]. Surplus dietary AA supplementation for GI microbiome modulation is a potential avenue to be explored for maintaining GI health and homeostasis [54]. For instance, Trp supplementation can increase populations of *Lactobacillus* and/or *Prevotella*, which are linked to short-chain fatty acid (SCFA) production, while reducing opportunistic pathogens such as *Clostridium* sp. and, in turn, improving gut barrier integrity and overall pig health [67,68]. More specifically, the GI microbial production of SCFAs by selective bacterial groups may improve epithelial cell proliferation and GI barrier function. As seen in the present study, which corroborates the findings of Pas et al. [56], sanitary conditions can affect the initial fecal microbial composition of pigs. However, this study suggests that the two treatment diets were similar enough in composition to approximate the HSS and LSS fecal microbiomes, suggesting that at the dose tested, the AA+ treatment did not cause major shifts in bacterial populations at the genus level. Instead, the alpha- and beta-diversity, co-occurrence network analysis, and data mining for identification of unique taxonomic changes revealed that over time, regardless of the treatment, a core microbiome composition and topology was achieved across all treatments (*Alloprevotella*, *Bifidobacterium*, *Blautia*, *Clostridium*, *Lactobacillus*, *Megasphaera*, *Prevotella*, *Ruminococcus*, *Streptococcus*, *Treponema*, etc.), as previously reported [44,69]. This absence of change in the major central taxa suggests a high resistance in the community structure. Obviously, 16S rRNA amplicon sequencing comes with caveats, such as not allowing for microbial absolute counts to be established and not providing enough taxonomic resolution for species-level annotation or strain identification [44,69,70,71]. Additionally, our fecal microbiome analysis was performed with a limited sample size. Nonetheless, surplus AA supplementation may not be consistent in changing the core GI microbiota composition at the bacterial genus/family level, but recent work by Beaumont et al. [26] suggests that further studies on microbial metabolic capabilities should be performed to understand the effect of these AAs, potentially including mycobiome data [72,73].

## 5. Conclusions

The results of this study partially support the hypothesis by demonstrating that supplementation with Trp, Thr, and Met improved the performance, health, and protein metabolism of healthy piglets raised under mixed management and poor housing conditions. On the other hand, it also demonstrated that healthy piglets raised under GOOD SC might not need surplus supplementation with AA for optimal growth, which in turn may lower the cost of production and facilitate management practices. Since piglets often become mixed and form heterogeneous batches in multiple-site production systems, our study suggests that increased Trp, Thr, and Met dietary supplementation could contribute to mitigating the side effects of this known risk factor in modern pig farms.

## Figures and Tables

**Figure 1 animals-14-01143-f001:**
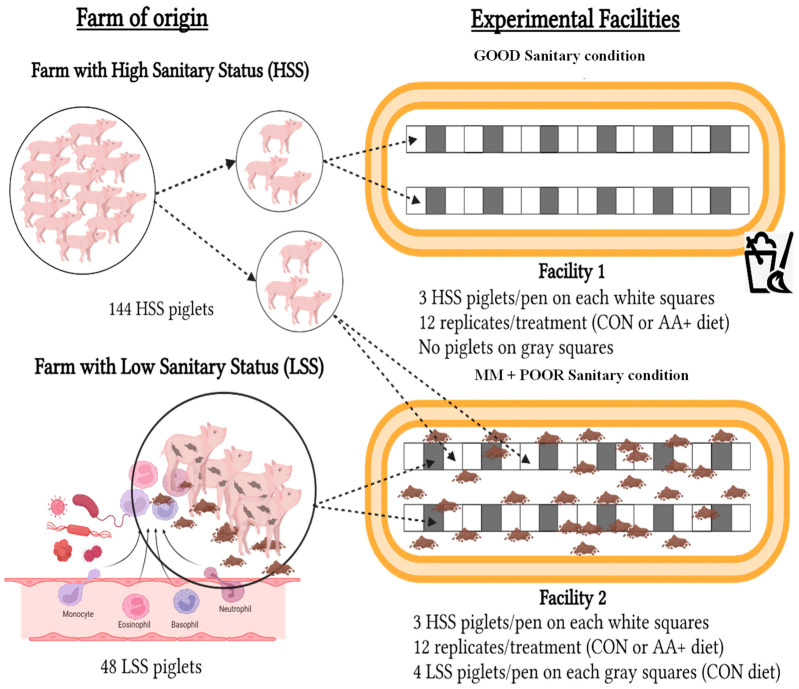
Experimental design presenting pen-based allocation of piglets across treatments. A total of 144 entire male piglets from a high-sanitary-status (HSS) multiplier farm were randomly distributed in a 2 × 2 factorial arrangement, composed of two sanitary conditions (Good or MM + Poor) and two diets (control (CON) or supplemented (AA+)). Each pen with HSS piglets was considered an experimental unit, except for the fecal microbiome analysis, where each piglet was considered an experimental unit. Good sanitary conditions were maintained by a daily cleaning protocol used to remove all manure from and under the slatted plastic floor. On the other hand, mixed management and poor sanitary conditions were mimicked by weekly cleaning of the facility while removing manure only from under the slatted plastic floor. Additionally, piglets from high- and low-sanitary-status (HSS and LSS, respectively) origins were co-housed to model poor sanitary conditions, where LSS piglets were non-experimental animals and were used only to compose the challenge model.

**Figure 2 animals-14-01143-f002:**
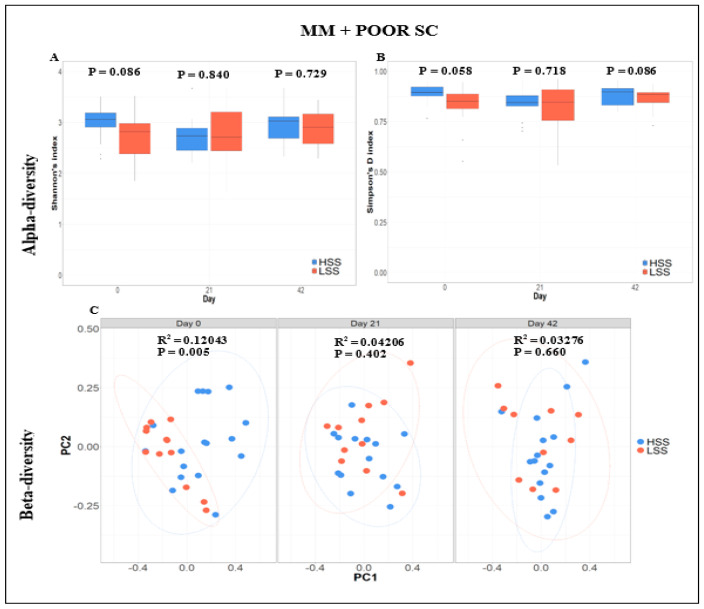
Alpha- and beta-diversity analysis of fecal microbiome samples across piglets of either high or low sanitary status (HSS vs. LSS), housed under mixed management and poor sanitary conditions (MM + Poor SC) regardless of the diet provided. (**A**) shows no significant difference in alpha-diversity (Shannon’s index) across HSS and LSS piglets over time under mixed management and poor sanitary conditions. (**B**) shows no significant difference in alpha-diversity (Simpson’s D Index) across HSS piglets over time under poor sanitary conditions. A Welch two-sample T-test was used to assess the difference between HSS and LSS piglets for alpha-diversity (Shannon’s and Simpson’s D indexes). (**C**) shows a significant difference in Bray–Curtis distance matrix across HSS and LSS piglets. Two principal components (PC1 and PC2) are shown for fecal samples across 0, 21, and 42 days of experiment. A PERMANOVA model was used to assess significant differences (*p* < 0.05), which were only found across HSS and LSS piglets at day 0 (*p* = 0.005, R^2^ = 0.12043). Beta-diversity differences across HSS and LSS piglets at arrival (d0) represent that farm sanitary conditions affect gastrointestinal microbiome composition. The sample sizes for the fecal samples were: MM + Poor SC (**A**–**C**) at day 0, HSS (n = 17) and LSS (n = 12); at day 21, HSS (n = 14) and LSS (n = 11); and day 42, HSS (n = 15) and LSS (n = 11). Each animal was considered an experimental unit throughout the analysis.

**Figure 3 animals-14-01143-f003:**
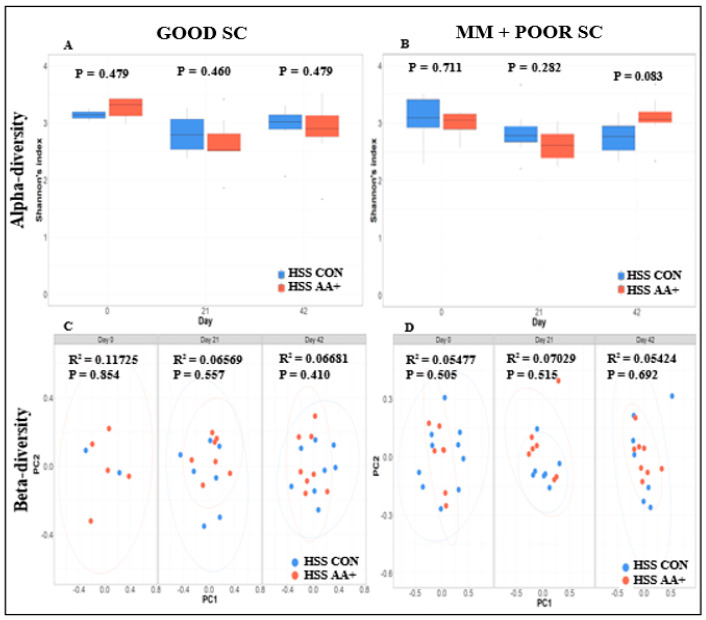
Alpha- and beta-diversity analysis of fecal microbiome samples across piglets with high sanitary status (HSS), housed in GOOD or MM + POOR sanitary conditions (SC), and fed with a control (CON) or extra-supplemented diet with amino acids (AA+). (**A**) shows no significant difference in alpha-diversity (Shannon’s Index) between diets (CON vs. AA+) across HSS piglets under GOOD SC over time. (**B**) shows no significant difference in alpha-diversity (Shannon’s Index) between diets (CON vs. AA+) across HSS piglets under MM + POOR SC over time. Alpha-diversity comparison between diet groups was conducted using a Welch two-sample T-test (*p* < 0.05). (**C**) shows no significant difference in Bray–Curtis distance matrix between diets across HSS piglets under GOOD SC over time (*p* < 0.05). (**D**) shows no significant difference in Bray–Curtis distance matrix between diets across HSS piglets under MM + POOR SC over time (*p* < 0.05). Two principal components (PC1 and PC2) are shown for fecal samples across 0, 21, and 42 days of experiment. A PERMANOVA model was used to assess diet effect in each sanitary condition (GOOD or MM + POOR) for beta-diversity (Bray–Curtis distances). The sample sizes for the fecal samples were: GOOD SC (**A**,**C**) at day 0, CON (n = 2) and AA+ (n = 5); at day 21, CON (n = 7) and AA+ (n = 7); and day 42, CON (n = 8) and AA+ (n = 8). MM+POOR SC (**B**,**D**) at day 0, HSS CON (n = 10), HSS AA+ (n = 7); at day 21, HSS CON (n = 7), HSS AA+ (n = 7); and day 42, HSS CON (n = 7), HSS AA+ (n = 8). Each piglet was considered an experimental unit throughout the analysis.

**Figure 4 animals-14-01143-f004:**
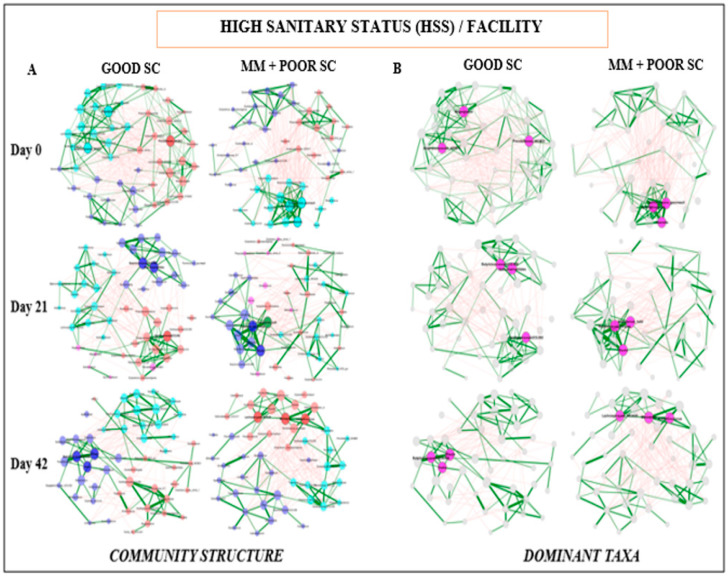
Co-occurrence network analysis of fecal microbiome samples across facilities (GOOD vs. MM+POOR SC) using only HSS animals. Each node represents a bacterial taxon, and the size of each node is scaled according to its centrality. Positive Spearman-based associations between taxa are depicted as green edges (the thicker the line, the stronger the association), whereas red edges are indicative of an anti-correlation between taxa. Edges representing values less than 0.5 are not shown. Taxa are colored based on clusters calculated in the network construction. (**A**) Comparison of co-occurrence networks between GOOD and MM+POOR SC over time. (**B**) Similarly, this plot represents the same correlation structure depicted in plot A, but only highlights the three most important taxa based on the highest degree of centrality (i.e., dominant taxa). The sample sizes for the fecal samples were: GOOD SC (**A**,**B**) at day 0 (n =7); at day 21 (n = 14); and day 42, CON (n = 16). MM+POOR SC (**A**,**B**) at day 0 (n = 17); at day 21 (n = 14); and at day 42 (n = 15). Each animal was considered an experimental unit throughout the analysis.

**Figure 5 animals-14-01143-f005:**
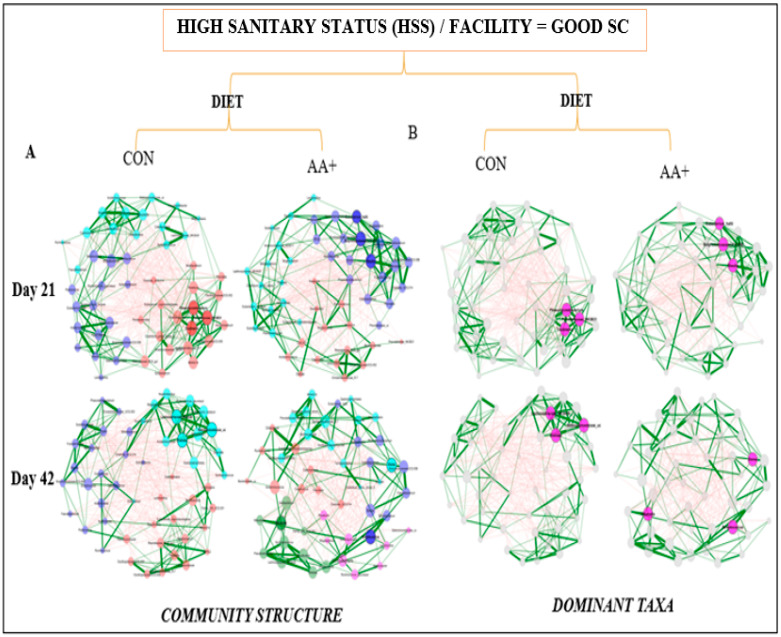
Temporal co-occurrence network analysis of fecal microbiome samples across barns using only HSS animals within the facility with GOOD SC while comparing across dietary treatments (CON for control, and AA+ for amino acid supplementation). Each node represents a bacterial taxon, and the size of each node is scaled according to its centrality. Positive Spearman-based associations between taxa are depicted as green edges (the thicker the line, the stronger the association), whereas red edges are indicative of an anti-correlation between taxa. Edges representing values less than 0.5 are not shown. Taxa are colored based on clusters calculated in the network construction. (**A**) Comparison of co-occurrence networks between CON vs. AA+ diets across days 21 and 42. (**B**) Similarly, this plot represents the same correlation structure depicted in plot A, but only highlights the three most important taxa based on the highest degree of centrality (i.e., dominant taxa). The sample sizes for the fecal samples were: GOOD SC (**A**,**B**) at day 21, CON (n = 7) and AA+ (n = 7); and day 42, CON (n = 8) and AA+ (n = 8). Day 0 was excluded from the analysis because of a small sample size. Each animal was considered an experimental unit throughout the analysis.

**Figure 6 animals-14-01143-f006:**
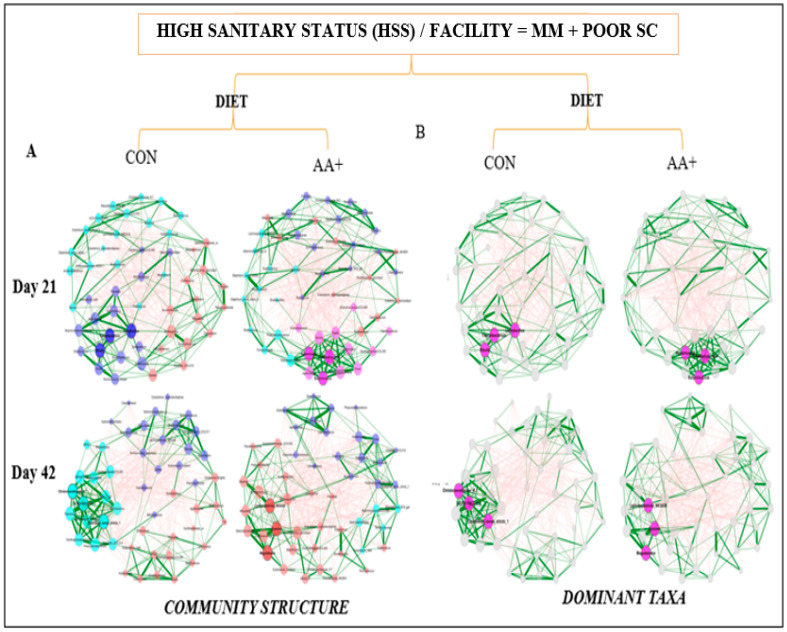
Temporal co-occurrence network analysis of fecal microbiome samples across barns using only HSS animals within the facility with MM+POOR SC while comparing across dietary treatments (CON for control, and AA+ for amino acid supplementation). Each node represents a bacterial taxon, and the size of each node is scaled according to its centrality. Positive Spearman-based associations between taxa are depicted as green edges (the thicker the line, the stronger the association), whereas red edges are indicative of an anti-correlation between taxa. Edges representing a value less than 0.5 are not shown. Taxa are colored based on clusters calculated in the network construction. (**A**) Comparison of co-occurrence networks between CON vs. AA+ diets across days 21 and 42. (**B**) Similarly, this plot represents the same correlation structure depicted in plot A, but only highlights the three most important taxa based on the highest degree of centrality (i.e., dominant taxa). The sample sizes for the fecal samples were: MM + POOR SC (A,B) at day 21, CON (n = 7) and AA+ (n = 7); and day 42, CON (n = 7) and AA+ (n = 8). Day 0 was excluded from the analysis because of a small sample size. Each animal was considered an experimental unit throughout the analysis.

**Figure 7 animals-14-01143-f007:**
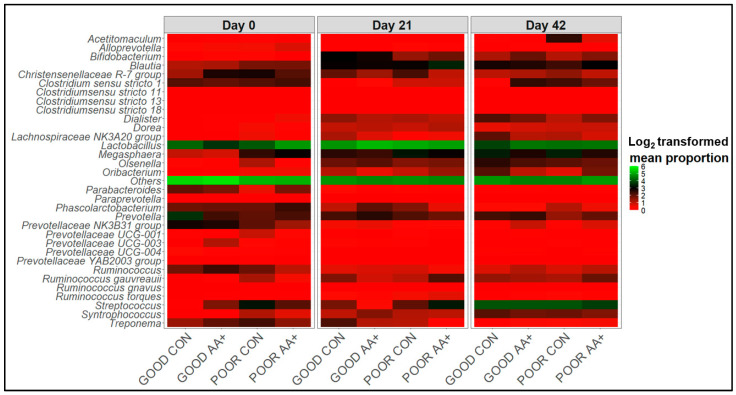
Temporal pattern of fecal microbiome taxa distribution across HSS animals by treatment. Log_2_-transformed mean proportion of keystone taxa across HSS animals over time across all treatments, which combined sanitary conditions (GOOD vs. MM + POOR SC) and diets (CON vs. AA+). All GOOD in the Figure refers to GOOD SC, while all POOR refers to MM + POOR SC. The most abundant (>2%) and most dominant taxa depicted across all network analyses were used to construct this heatmap distribution of keystone taxa classified at either the family, genus, or species levels. The stronger the color, the more abundant a given taxon is, on average, across treatments, and over time. The sample sizes for the fecal samples were: GOOD SC at day 0, GOOD CON (n = 2) and GOOD AA+ (n = 5); at day 21, GOOD CON (n = 7) and GOOD AA+ (n = 7); and day 42, GOOD CON (n = 8) and GOOD AA+ (n = 8). MM + POOR SC at day 0, POOR CON (n = 10), POOR AA+ (n = 7); at day 21, POOR CON (n = 7), POOR AA+ (n = 7); and day 42, POOR CON (n = 7), POOR AA+ (n = 8). Each animal was considered an experimental unit throughout the analysis.

**Table 1 animals-14-01143-t001:** Ingredients and nutrient compositions of the experimental diets.

Ingredients, %	Phase 1 (d 0 to 14)		Phase 2 (d 15 to 28)		Phase 3 (d 29 to 42)	
	CON ^1^	AA+ ^1^	CON	AA+	CON	AA+
Corn	49.41	49.41	58.85	58.85	69.64	69.64
Soybean meal	15.00	15.00	20.00	20.00	26.00	26.00
Whey protein	21.42	21.42	12.14	12.14	-	-
Soy protein concentrated (60% CP)	5.23	5.23	2.55	2.55	-	-
Spray-dried blood plasma	5.00	5.00	2.50	2.50	-	-
Dicalcium phosphate	0.870	0.870	0.655	0.655	0.512	0.512
Limestone	0.795	0.795	0.884	0.884	0.819	0.819
Soybean oil	-	-	0.100	0.100	0.125	0.125
Salt	0.330	0.330	0.445	0.445	0.450	0.450
Mineral premix ^2^	0.110	0.110	0.100	0.100	0.090	0.090
Vitamin premix ^3^	0.060	0.060	0.050	0.050	0.050	0.050
Choline chloride	0.060	0.060	0.060	0.060	0.060	0.060
Corn starch	0.500	0.105	0.500	0.146	1.000	0.665
Sodium bicarbonate	-	-	-	-	0.053	0.053
Enzae^®^ (phytase) ^4^	0.005	0.005	0.005	0.005	0.005	0.005
Biolys (54.6%) ^5^	0.66	0.66	0.68	0.68	0.77	0.77
DL-Met (99%) ^5^	0.23	0.39	0.19	0.34	0.16	0.30
L-Thr (98.5%) ^5^	0.14	0.32	0.13	0.30	0.15	0.30
L-Trp (98%) ^5^	0.001	0.05	0.008	0.04	0.008	0.04
L-Val (98%) ^5^	0.11	0.11	0.11	0.11	0.09	0.09
L-Ile	0.04	0.04	0.01	0.01	-	-
Total	100	100	100	100	100	100
Nutrients, calculated values						
ME, kcal/kg	3386	3391	3354	3358	3331	3335
NE, kcal/kg	2521	2524	2516	2518	2470	2473
CP, %	21.2	21.5	19.5	19.9	18.8	18.98
SID Lys, %	1.50	1.50	1.35	1.35	1.23	1.23
SID Met + Cys, %	0.82	0.98	0.74	0.89	0.68	0.82
SID Met, %	0.48	0.64	0.44	0.59	0.41	0.55
SID Thr, %	0.88	1.06	0.79	0.95	0.73	0.88
SID Trp, %	0.25	0.30	0.22	0.26	0.20	0.24
SID Val, %	1.02	1.02	0.92	0.92	0.84	0.84
SID Arg, %	1.11	1.11	1.05	1.05	1.08	1.08
SID Ile, %	0.80	0.80	0.72	0.72	0.69	0.69
SID Leu, %	1.58	1.58	1.46	1.46	1.42	1.42
SID His, %	0.52	0.52	0.46	0.46	0.44	0.44
SID Phe + Tyr, %	1.61	1.61	1.45	1.45	1.38	1.38
SID Phe, %	0.94	0.94	0.85	0.85	0.81	0.81
STTD P, %	0.45	0.45	0.40	0.40	0.33	0.33
Ca, %	0.85	0.85	0.80	0.80	0.70	0.70
Na, %	0.40	0.40	0.35	0.35	0.20	0.20
Lactose, %	15.00	15.00	8.50	8.50	-	-
Nutrients, total analyzed values						
CP, %	21.6	22.0	20.7	21.0	19.2	19.3
Lys, %	1.58	1.59	1.50	1.58	1.36	1.38
Met + Cys, %	0.89	0.94	0.85	0.99	0.74	0.81
Met, %	0.48	0.53	0.48	0.63	0.43	0.51
Thr, %	1.02	1.08	0.91	1.05	0.84	0.90
Val, %	1.16	1.20	1.10	1.09	0.96	0.94
Arg, %	1.23	1.25	1.20	1.22	1.15	1.13
Ile, %	0.84	0.85	0.80	0.82	0.77	0.76
Leu, %	1.79	1.81	1.72	1.73	1.66	1.70
His, %	0.56	0.56	0.53	0.51	0.50	0.50
Phe, %	1.01	1.03	0.95	0.96	0.89	0.90

^1^ CON, AA profile diet according to the NRC (2012); AA+, diet with Thr:Lys, Met+Cys:Lys, and Trp:Lys ratios increased by 20% above the CON diet. ^2^ Mineral supplement (per kg of feed): manganese 0.0522 g; zinc 0.1375 g; iron 0.1045 g; copper 0.016 g; iodine 1.2 mg; selenium 0.121 mg. ^3^ Vitamin supplement (per kg of feed): vitamin A 20600000 IU; 25-Hydroxy Vitamin D3 4600000 IU; vitamin E 130000 IU; vitamin K3 0.054 g; vitamin B1 0.0036 g; vitamin B2 0.012 g; vitamin B6 0.036 g; folic acid 0.00204 g, biotin 0.000192 g; niacin 0.054 g; vitamin B5 0.03 g. ^4^ Phytase was provided by Cargill and contained 500 FTU/ton. ^5^ Amino acids (BioLys, MetAmino, ThreAmino, TrypAmino, ValAmino, respectively) were provided by Evonik Nutrition & Care GmbH (Hanau-Wolfgang, Germany).

**Table 2 animals-14-01143-t002:** Serum white blood cell (WBC) counts in piglets from farms with high or low sanitary status (HSS or LSS).

Item (no. × 10^9^/L)	HSS Piglets	LSS Piglets	RSD ^1^	*p*-Value
Leukocytes	17.2	20.6	0.349	<0.01
Granulocytes ^2^	7.7	9.9	0.749	<0.01
Lymphocytes	8.5	9.4	0.422	0.19
Monocytes	1.0	1.3	0.053	<0.01

^1^ Residual standard deviation; ^2^ granulocytes correspond to the sum of neutrophils, eosinophils, and basophils. Each pen was considered an experimental unit (HSS piglets n = 12; LSS piglets n = 12).

**Table 3 animals-14-01143-t003:** Performance of piglets fed control or amino-acid-supplemented diet above the requirement, raised under good or mixed management and poor sanitary conditions (SC) during the nursery phase (42 days).

	GOOD SC ^1^			MM + POOR SC ^1^				*p*-Value
Item	CON ^1^	AA+ ^1^	*p* Value	CON	AA+	*p* Value	RSD ^2^	SC
Initial BW, kg	6.35	6.26	0.64	6.26	6.16	0.57	0.215	0.51
Phase 1 (d 0 to 14)								
Final BW, kg	8.91	8.85	0.83	8.24	8.61	0.17	0.313	0.03
ADFI, g	282	256	0.19	245	223	0.45	32	0.03
ADG, g	182	185	0.89	123	154	0.32	37	0.02
G:F	0.67	0.72	0.70	0.49	0.65	0.19	0.138	0.15
Phase 2 (d 15 to 28)								
Final BW, kg	15.60	16.12	0.45	12.76	13.86	0.09	0.723	<0.01
ADFI, g	595	646	0.35	434	523	0.06	55	<0.01
ADG, g	482	524	0.22	325	373	0.10	46	<0.01
G:F	0.84	0.83	0.91	0.77	0.77	1.00	0.078	0.16
Phase 3 (d 29 to 42)								
Final BW, kg	22.71	23.85	0.26	18.20	20.95	<0.01	1.053	<0.01
ADFI, g	834	953	0.02	652	824	<0.01	64	<0.01
ADG, g	503	555	0.24	395	512	<0.01	46	<0.01
G:F	0.61	0.58	0.34	0.61	0.61	0.87	0.031	0.68
Overall (d 0 to 42)								
ADFI, g	573	614	0.17	441	523	0.01	38	<0.01
ADG, g	394	425	0.32	282	345	0.01	37	<0.01
G:F	0.69	0.69	0.86	0.63	0.65	0.34	0.034	0.01

^1^ CON, AA profile diet according to the NRC (2012) [27]; AA+, diet with Thr:Lys, Met+Cys:Lys, and Trp:Lys ratios increased by 20% above the CON diet; GOOD SC, facility without mixing piglets from different origins and daily cleaning; MM + POOR SC, facility with a mixed management of piglets from different origins and poor sanitary conditions. ^2^ Residual standard deviation. Each pen was considered an experimental unit (n = 12).

**Table 4 animals-14-01143-t004:** Serum concentration of acute-phase proteins of piglets fed control or amino-acid-supplemented diet above the requirement raised under good or mixed management and poor sanitary conditions (SCs) during the nursery phase (42 days).

Item	GOOD SC ^1^			MM + POOR SC ^1^				*p*-Value
	CON ^1^	AA+ ^1^	*p*-Value	CON	AA+	*p*-Value	RSD ^2^	SC
Day 0								
IgA, mg/mL	1.50	1.40	0.51	1.30	1.50	0.36	0.024	0.68
IgG, mg/mL	5.90	6.10	0.76	5.10	6.40	0.13	0.096	0.62
Ceruloplasmin, mg/mL	0.90	0.90	0.95	0.80	1.00	0.15	0.013	0.45
Transferrin, mg/mL	3.60	3.90	0.29	3.50	3.50	0.99	0.031	0.15
Albumin, g/L	35.3	35.1	0.94	34.9	34.5	0.85	0.238	0.75
Haptoglobin, mg/mL	0.50	0.60	0.73	0.60	0.70	0.22	0.014	0.19
α-1 acid glycoprotein, µg/mL	38.6	48.6	0.84	45.7	51.4	0.69	0.013	0.53
Day 21								
IgA, mg/mL	1.30	1.20	0.29	1.40	1.20	0.17	0.025	0.73
IgG, mg/mL	6.90	6.60	0.82	8.40	7.20	0.24	0.106	0.16
Ceruloplasmin, mg/mL	0.60	0.60	0.54	0.70	0.70	0.77	0.014	0.31
Transferrin, mg/mL	3.80	3.90	0.84	3.70	4.10	0.26	0.044	0.91
Albumin, g/L	28.8	28.8	0.99	28.4	29.5	0.53	0.197	0.92
Haptoglobin, mg/mL	0.40	0.30	0.52	1.00	0.60	0.07	0.025	0.002
α-1 acid glycoprotein, µg/mL	45.0	38.3	0.44	48.6	51.4	0.29	0.019	0.13
Day 42								
IgA, mg/mL	1.50	1.30	0.41	1.40	1.40	0.80	0.025	0.89
IgG, mg/mL	9.50	9.80	0.87	10.40	12.30	0.29	0.185	0.18
Ceruloplasmin, mg/mL	1.10	0.90	0.20	0.80	1.00	0.18	0.014	0.13
Transferrin, mg/mL	4.30	4.60	0.21	4.20	4.30	0.82	0.026	0.44
Albumin, g/L	33.5	34.1	0.78	31.3	31.1	0.91	0.187	0.05
Haptoglobin, mg/mL	1.00	0.70	0.16	0.60	0.80	0.35	0.022	0.43
α-1 acid glycoprotein, µg/mL	48.3	48.6	0.94	50.0	34.3	0.14	0.014	0.38

^1^ CON, AA profile diet according to the NRC (2012); AA+, diet with Thr:Lys, Met+Cys:Lys, and Trp:Lys ratios increased by 20% above the CON diet; GOOD SCs, facility without mixing piglets from different origins and daily cleaning; MM + POOR SCs, facility with a mixed management of piglets from different origins and poor sanitary conditions. ^2^ Residual standard deviation. Each pen was considered an experimental unit (n = 12).

**Table 5 animals-14-01143-t005:** Serum biochemical profile of piglets fed control or amino-acid-supplemented diet above the requirement and raised under good or mixed management and poor sanitary conditions (SC) during the nursery phase (42 days).

Item	GOOD SC ^1^			MM + POOR SC ^1^				*p*-Value
	CON ^1^	AA+ ^1^	*p*-Value	CON	AA+	*p*-Value	RSD ^2^	SC
Day 0								
Total protein, g/dL	5.22	5.21	0.95	5.01	5.09	0.76	0.277	0.39
Urea, mg/dL	17.96	19.44	0.66	24.77	18.60	0.10	3.520	0.24
Creatinine, mg/dL	1.59	1.44	0.10	1.65	1.46	0.05	0.090	0.51
Lactate, mg/dL	50.57	59.38	0.26	43.07	31.13	0.13	7.710	0.003
Lactate dehydrogenase, U/L	1415	1345	0.33	1485	1303	0.01	69.980	0.78
Day 21								
Total protein, g/dL	4.68	4.65	0.91	4.78	4.84	0.824	0.272	0.45
Urea, mg/dL	19.35	19.18	0.96	27.13	16.35	0.003	3.420	0.30
Creatinine, mg/dL	1.08	1.08	0.97	1.19	1.04	0.04	0.070	0.42
Lactate, mg/dL	48.79	51.25	0.75	41.21	44.19	0.70	7.720	0.18
Lactate dehydrogenase, U/L	1333	1314	0.80	1227	1279	0.49	76.810	0.18
Day 42								
Total protein, g/dL	5.59	5.69	0.69	5.45	5.63	0.43	0.235	0.54
Urea, mg/dL	23.38	18.67	0.16	26.06	19.21	0.04	3.310	0.49
Creatinine, mg/dL	1.31	1.34	0.65	1.27	1.22	0.47	0.080	0.14
Lactate, mg/dL	67.28	71.09	0.81	85.75	48.71	0.03	15.990	0.86
Lactate dehydrogenase, U/L	1256	1403	0.09	1281	1253	0.75	87.250	0.31

^1^ CON, AA profile diet according to the NRC (2012); AA+, diet with Thr:Lys, Met+Cys:Lys, and Trp:Lys ratios increased by 20% above the CON diet; GOOD SC, facility without mixing piglets from different origins and daily cleaning; MM + POOR SC, facility with mixed management of piglets from different origins and poor sanitary condition. ^2^ Residual standard deviation.

**Table 6 animals-14-01143-t006:** Plasma redox parameters of piglets fed control or amino-acid-supplemented diet above the requirement and raised under GOOD or MM + POOR sanitary conditions (SC) during the nursery phase (42 days).

Item ^3^	GOOD SC ^1^			MM + POOR SC ^1^				*p*-Value
	CON ^1^	AA+ ^1^	*p* Value	CON	AA+	*p* Value	RSD ^2^	SC
GSH, pmol.mL^−1^	20.27	22.47	0.46	19.57	22.99	0.31	2.32	0.96
GSSG, pmol.mL^−1^	8.02	5.92	0.16	5.94	6.89	0.62	1.24	0.64
GSH:GSSG E_h_, mV ^4^	−316.9	−324.3	0.05	−320.5	−321.4	0.55	3.139	0.89
GST, mU.mL^−1^	6.82	6.94	0.82	7.21	7.30	0.85	0.42	0.32
LPO, mmol.mL^−1^	38.75	25.17	0.18	32.79	28.64	0.69	7.28	0.70
SOD, U.mL^−1^	47.02	46.79	0.91	44.04	44.16	0.90	1.15	0.01
ROS, DCFHA-DA ^5^	458.8	336.9	0.21	601.7	440.6	0.27	88.26	0.08
T-AOC, µM.L^−1^	2.99	3.51	0.80	1.55	2.61	0.49	1.97	0.29

^1^ CON, AA profile diet according to the NRC (2012); AA+, diet with Thr:Lys, Met+Cys:Lys, and Trp:Lys ratios increased by 20% above the CON diet; GOOD SC, facility without mixing piglets from different origins and daily cleaning; MM+POOR SC, facility with a mix of piglets from different origins and poor hygiene conditions. ^2^ Residual standard deviation. ^3^ ROS: reactive oxygen species; LPO: lipid peroxidation; T-AOC: total antioxidant capacity; SOD: superoxide dismutase; GST: glutathione S-transferase; GSH: reduced glutathione; GSSH: glutathione disulfide, and GSH:GSSH ratio. ^4^ Nernst equation (E_h_ = −264 − (61.5/2) × log GSH^2^/GSSG) was applied to determine the redox potential of GSH:GSSG ratio. ^5^ DCFHA-DA: ROS detection by 2′,7′-dichlorofluorescein diacetate probes emitting fluorescence due to probe oxidation.

## Data Availability

All R code and tabular data for microbiome analysis can be found in the following Figshare link: https://figshare.com/projects/Weaned_piglets_under_poor_sanitary_condition/162358, accessed on 17 March 2023. All microbiome 16S rRNA amplicon sequencing data can be found here https://www.ncbi.nlm.nih.gov/bioproject/PRJNA985664, accessed on 20 June 2023. Other data are contained within the article.

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
