# Peer review of "Increased Dietary Trp, Thr, and Met Supplementation Improves Performance, Health, and Protein Metabolism of Weaned Piglets under Mixed Management and Poor Housing Conditions"

_animals, 2024, doi:10.3390/ani14081143_

Round 1

Reviewer 1 Report

Comments and Suggestions for Authors

1. Please provide the basic information for the 16S rRNA gene sequencing, e,g, Total reads, average reads per sample. These data are essential to prove that the sequencing quality is good enough to do further analysis;

2. Line 224, "2.3.5.16. S rRNA" please revise;

3. Line 231,  "then rarefied to 5,000 reads per sample", why use 5000 reads per sample, is it enough for the un-biased analysis, please explain;

4. Table 5., did you analysis concentrations of free amino acids (e.g., Met, Trp, Thr, Gly, Glu, Gln, Cys) in the blood of piglets. These data will support your findings;

5. Figure 7. please change the color profiles of the heatmap to red-green-black;

6. The two diets you compared were not isonitrogenous, please explain the reason.

Author Response

We would like thank you for your suggestions. 

  1. Please provide the basic information for the 16S rRNA gene sequencing, e,g, Total reads, average reads per sample. These data are essential to prove that the sequencing quality is good enough to do further analysis.

Total reads are 9111100 and average reads number per sample is 26032.

  1. Line 224, "2.3.5.16. S rRNA" please revise.

It was done in the manuscript.

  1. Line 231, "then rarefied to 5,000 reads per sample", why use 5000 reads per sample, is it enough for the un-biased analysis, please explain.

The 5,000 reads were chosen because of the distribution of the sequences per sample ranging from 3,000 to 80,000. A higher number of reads could result in loss of read samples, affecting the sample number for statistical analysis. Considering fecal samples of pigs, the 5,000 reads had been used also by others, as may be viewed in the following publications.

https://www.ncbi.nlm.nih.gov/pmc/articles/PMC6606507/

https://www.ncbi.nlm.nih.gov/pmc/articles/PMC7356342/

  1. Table 5., did you analysis concentrations of free amino acids (e.g., Met, Trp, Thr, Gly, Glu, Gln, Cys) in the blood of piglets. These data will support your findings.

Unfortunately, due to financial constraints, we did not evaluate the free amino acids concentration in the blood of piglets. Although these data could have supported our findings, we believe their absence does not invalidate the inferences from our results and our conclusion.

  1. Figure 7. please change the color profiles of the heatmap to red-green-black.

We changed the color of the Figure 7 as suggested.

  1. The two diets you compared were not isonitrogenous, please explain the reason.

The differences in nitrogen content between diets in each phase were very low, approximately 0.04%, which was expected, as including more amino acids results in an increase in nitrogen levels. However, since all other nitrogen sources were included in equal amounts in both diets, the disparity in nitrogen levels stems from the inclusion of the three tested amino acids (Met, Thr, and Trp). Therefore, we assume that the dietary effects observed in our study stem from the amino acid supplementation and their functional action and not due to the low N difference between diets.

Reviewer 2 Report

Comments and Suggestions for Authors

This manuscript mainly studied the effect of amino acids supplementation, particularly Trp, Thr, and Met, on the performance of piglets with different housing conditions. It found that AA supplementation increased the growth performance of piglets under low sanitary conditions. The study provides a new nutritional strategy to handle the piglets under low sanitary conditions. This reviewer has the following comments:

1. Why did the authors choose 20% higher Trp, Thr, and Met in the study?

2. The nutrition level of the diets such as CP, Ca, P, and AA need provide the analyzed values.

3. The increased ADG was mainly caused by increased feed intake, what’s the mechanism for the improvement of feed intake besides Trp? Is the diarrhea rate being affected?

4. Sanitary condition has a significant effect on the number of white blood cells (WBC), which indicates WBC is a good marker for the status of piglets under different raising conditions. Does AA supplementation affect the number of WBC?

5. Suggest the unit for ADG and ADFI uses g instead of kg in Table 3.

Author Response

We would like thank you for your suggestions. 

  1. Why did the authors choose 20% higher Trp, Thr, and Met in the study?

We chose the increase of 20% based on other papers published that observed that a supplementation 20% higher of these three amino acids had beneficial effects for challenged pigs. Although in those studies the challenge model differed from our model we used them as reference.

References:

Rodrigues, L.A.; Wellington, M.O.; González-Vega, J.C.; Htoo, J.K.; Kessel, A.Gv.; Columbus, D.A. A longer adaptation period to a functional amino acid-supplemented diet improves growth performance and immune status of Salmonella Typhimurium-challenged pigs. J. Anim. Sci. 2021, 99, skab146. 10.1093/jas/skab146

Rodrigues, L.A.; Wellington, M.O.; González-Vega, J.C.; Htoo, J.K.; Kessel, A.Gv.; Columbus, D.A. Functional amino acid supplementation, regardless of dietary protein content, improves growth performance and immune status of weaned pigs challenged with Salmonella Typhimurium. J. Anim. Sci. 2021, 99, skaa365. 10.1093/jas/skaa365

Valini, G.A.C.; Arnaut, P.R.; França, I.; Ortiz, M.T.; de Oliveira, M.J.K.; Melo, A.D.B.; Marçal, D.A.; Campos, P.H.R.F.; Htoo, J.K.; Brand, H.G.; Hauschild, L. Increased dietary Trp, Thr, and Met supplementation improves growth performance and protein deposition of Salmonella-challenged growing pigs under poor housing conditions. J. Anim. Sci. 2023, 101, 1–12. 10.1093/jas/skad141

  1. The nutrition level of the diets such as CP, Ca, P, and AA need provide the analyzed values.

Total analyzed CP and AA content, except Trp, were included between parentheses in diet composition table. We did not analyze Ca and P because those nutrients were not the subject of the study. Furthermore, based on dicalcium phosphate and limestone supplementation it was not expected to have deficiency of Ca and P.

  1. The increased ADG was mainly caused by increased feed intake, what’s the mechanism for the improvement of feed intake besides Trp? Is the diarrhea rate being affected?

We revised this sentence in the text based on Zhang et al., 2007 (Lines 528-533). Unfortunately, fecal consistency was not evaluated in our study, although a possible data of diarrhea rate being reduced by AA supplementation could support our findings for growth performance.

  1. Sanitary condition has a significant effect on the number of white blood cells (WBC), which indicates WBC is a good marker for the status of piglets under different raising conditions. Does AA supplementation affect the number of WBC?

The number of WBC was not evaluated after the beginning of the sanitary challenge. We focused our hypothesis on serum Haptoglobin levels since moderate increase of this marker level is highly correlated to decreased ADG under poor sanitary condition (Le Floc´h et al., 2021). Based on our study, we cannot affirm if AA supplementation affect the number of WBC. However, a diet formulated to be marginally deficient at 95% of the Lys and deficient in Met + Cys, Thr, and Trp at 75% of their requirement (according to Central Bureaux for Livestock Feeding), did not affect WBC of piglets under contrasting sanitary challenge (Hoek et al., 2016).

  1. Suggest the unit for ADG and ADFI uses g instead of kg in Table 3.

Table 3 was modified in the manuscript as suggested.

Reviewer 3 Report

Comments and Suggestions for Authors

The objective of the study was to evaluate the effect of diet supplementation with mixture of three amino acids (tryptophan, threonine and methionine) on growth performance, metabolism, immunological and oxidative status and fecal microbiota composition in piglets blood parameters, fecal metabolites and microbiom in piglets after weeaning.. That manuscript is in scope of journal, however it needs many changes and explanation before acceptation for publication. Below you can find my suggestion for consideration:

1.      I suggest to modify the title to have the information on other evaluated parameters than growth performance

2.      Why the diets were supplemented with mixtures of three amino acids, but not single amino acids. Now Authors do not know, which amino acid is responsible for obtained effects.  Content of three amino acids has been increased by 20%. Why not more or less? Both deficiency and excess of dietary amino acids may have negative influence on growth and metabolism of animals.

3.      Threonine, methionine and tryptopfan are amino acids which play the important role in many physiological functions. Please, try explain why amino acids supplementation infuenced only growth performance, but not metabolism, imunological and oxidative status and fecal microbiota composition.

4.      It is mentioned in statistical analyses section that two-way ANOVA has been used. Where are results of statistical evaluation? Statistical method chosen by Authors didi nt give to possibility to get information if AA suplementation of diet improve the results of MM+POOR S.C. to the levels obtained in GOOD SC. It is not new that MM+POOR SC pigs have lower parameters than GOOD SC pigs.

5.      Please, correct the conclusion because obtained results do not suport the hypothesis which was that Thr + Met + Trp supplementation affect growth performance, metabolism, immunological and and oxidative status and fecal microbiota of piglets rainsed in different sanitary conditio

6.      In my opinion hypotesis should be modified. I suggest: possibility of improving growth and health of piglets rainsed in poor sanitary conditio by dietary AA supplementation to the level obtained in piglets rainsed in good sanitary conditions.

Author Response

We would like thank you for your suggestions.

  1. I suggest to modify the title to have the information on other evaluated parameters than growth performance

We revised the title as suggested.

  1. Why the diets were supplemented with mixtures of three amino acids, but not single amino acids. Now Authors do not know, which amino acid is responsible for obtained effects.  Content of three amino acids has been increased by 20%. Why not more or less? Both deficiency and excess of dietary amino acids may have negative influence on growth and metabolism of animals.

Diets were supplemented using these three amino acids considering their different pathways of action for pigs with immune system activation, which may result in a greater contribution to support the health of challenged pigs. Thus, this manuscript aimed to discuss the possible contributions of this supplementation based on carried out analyses and results found in the literature. In this study, we increased these amino acids 20% higher than NRC base on the beneficial effects observed in the literature supporting health for challenged pigs. Since the present study observed that increasing these amino acids in 20% presented beneficial effects, further studies may investigate if lower or higher levels could be more or less effective.

  1. Threonine, methionine and tryptophan are amino acids which play the important role in many physiological functions. Please, try explain why amino acids supplementation influenced only growth performance, but not metabolism, immunological and oxidative status and fecal microbiota composition.

We have reported higher haptoglobin levels, lower albumin synthesis, and a tendency for higher circulating ROS in piglets under MM+POOR SC as changes in immune and oxidative parameters. As such, the AA+ diet tended to reduce haptoglobin levels in piglets under MM+POOR SC, which may, in part, explain the supplementation effect in attenuating POOR SC pigs’ reduced growth performance (Lines 579-581). Furthermore, piglets under MM+POOR SC fed AA+ had lower serum concentrations of urea (on Days 21 and 42) and creatinine (on Day 21) than those in CON diet piglets, which indicate an improvement in piglet´s protein metabolism (Lines 585-587). Probably, blood collection after 21 or 42 days after the beginning of the proposed sanitary challenge were not the most adequate moments to find a most prominent effect of the AA+ diet on oxidative parameters. A possible explanation for the absent effect of diet on fecal microbiota composition was explained between Lines 612 and 615.

  1. It is mentioned in statistical analyses section that two-way ANOVA has been used. Where are results of statistical evaluation? Statistical method chosen by Authors did not give to possibility to get information if AA supplementation of diet improve the results of MM+POOR SC to the levels obtained in GOOD SC. It is not new that MM+POOR SC pigs have lower parameters than GOOD SC pigs.

P-values from ANOVA were reported in the text. Since no significant interactions (P<0.05) were observed and mainly sanitary condition effects were observed, we chose show ANOVA P-values in the text. Additionally, to further explore our data, we analyzed the effects of amino acid supplementation within each sanitary condition and presented these results in the tables.

  1. Please, correct the conclusion because obtained results do not support the hypothesis which was that Thr + Met + Trp supplementation affect growth performance, metabolism, immunological and oxidative status, and fecal microbiota of piglets raised in different sanitary condition.

We changed the conclusion now affirming that our results partially support the hypothesis by demonstrating that supplementation with Trp, Thr and Met improved the performance, health and protein metabolism of healthy piglets raised under mixed management and poor housing conditions.

  1. In my opinion hypothesis should be modified. I suggest possibility of improving growth and health of piglets raised in poor sanitary condition by dietary AA supplementation to the level obtained in piglets raised in good sanitary conditions.

We cannot accept this suggestion since we not expected similar performance between GOOD and POOR SC piglets, regardless of diet provided. We hypothesized an improvement on performance, health, and metabolism in AA+ piglets compared to CON piglets under POOR SC. Since interactions between Diet and SC were not observed in the present study, we evaluate the effect of diet separately in each SC. A similar model was also used by Rodrigues et al. (2021), where they had a 2×2×2 factorial design being one of the factors the sanitary challenge. These authors analyzed the triple factorial design and also examined the dietary effects within the challenged pigs. In their study, two factors related to the diets were considered: crude protein (CP) level and amino acid (AA) level.

Reviewer 4 Report

Comments and Suggestions for Authors

The authors evaluated the effect of amino acids supplementation on piglet responses under suboptimal herd health induced by a sanitary challenge. This study suggests that increased Trp, Thr and Met (AA+) dietary supplementation could contribute to mitigating the side effects of the harmful risk factor in modern pig farms. The manuscript is well written and the topic is sound. However, there are some issues such as statistical analysis need to be solved.  

1.     I do know if there could be so many Co-first authors.

2.     The authors should provide the reasons why used Thr, Met and Trp as the AA solution and how to set the ratio of each AA in this study.

3.     The statistical analysis in the whole manuscript should be modified. Since the experimental design is a 2 • 2 factorial design, thus I think that the authors could not analyze the dietary separately within each SC but include main effects (AA and SC) and interaction effect (AA • SC).

Author Response

We would like thank you for your suggestions.

  1. I do know if there could be so many Co-first authors.

It was changed based on Animals MDPI rules allowing only two co-first authors.

  1. The authors should provide the reasons why used Thr, Met and Trp as the AA solution and how to set the ratio of each AA in this study.

Diets were supplemented using these three amino acids considering their different pathways of action, which may result in a greater contribution to support the health of challenged pigs. Thus, this manuscript aimed to discuss the possible contributions of this supplementation based on carried out analyses and results found in the literature. We chose the increase of 20% based on other papers published that observed that a supplementation 20% higher of these three amino acids had beneficial effects for challenged pigs. Although in those studies the challenge model differed from our model we used them as reference.

References:

Rodrigues, L.A.; Wellington, M.O.; González-Vega, J.C.; Htoo, J.K.; Kessel, A.Gv.; Columbus, D.A. A longer adaptation period to a functional amino acid-supplemented diet improves growth performance and immune status of Salmonella Typhimurium-challenged pigs. J. Anim. Sci. 2021, 99, skab146. 10.1093/jas/skab146

Rodrigues, L.A.; Wellington, M.O.; González-Vega, J.C.; Htoo, J.K.; Kessel, A.Gv.; Columbus, D.A. Functional amino acid supplementation, regardless of dietary protein content, improves growth performance and immune status of weaned pigs challenged with Salmonella Typhimurium. J. Anim. Sci. 2021, 99, skaa365. 10.1093/jas/skaa365

Valini, G.A.C.; Arnaut, P.R.; França, I.; Ortiz, M.T.; de Oliveira, M.J.K.; Melo, A.D.B.; Marçal, D.A.; Campos, P.H.R.F.; Htoo, J.K.; Brand, H.G.; Hauschild, L. Increased dietary Trp, Thr, and Met supplementation improves growth performance and protein deposition of Salmonella-challenged growing pigs under poor housing conditions. J. Anim. Sci. 2023, 101, 1–12. 10.1093/jas/skad141

  1. The statistical analysis in the whole manuscript should be modified. Since the experimental design is a 2 • 2 factorial design, thus I think that the authors could not analyze the dietary separately within each SC but include main effects (AA and SC) and interaction effect (AA • SC).

We presented the effects of the 2x2 factorial analysis in the text. Since interactions between Diet and SC were not observed in the present study, we evaluate the effect of diet separately in each SC. A similar model was also used by Rodrigues et al. (2021), where they had a 2×2×2 factorial design being one of the factors the sanitary challenge. These authors analyzed the triple factorial design and also examined the dietary effects within the challenged pigs. In their study, two factors related to the diets were considered: crude protein (CP) level and amino acid (AA) level.

Round 2

Reviewer 1 Report

Comments and Suggestions for Authors

1. Please specify the total numbers of samples you have sequenced for microbiota analysis in the Materials and methods. Please provide the Summary of detailed information for reads of samples of each group as Supplemental Table 2.

Author Response

Dear Reviewer, thank you for all comments to improve our manuscript. 

Please, find below our responses:

L231-233 (M&M): A total of 3,384,304 sequences were reads with an average of 25,638 reads per sample for the bacterial 16S rRNA analysis (Table S1).

Reviewer 2 Report

Comments and Suggestions for Authors

1. It should include the reason for choosing 20% higher Trp, Thr, and Met in the manuscript.

2. SID AA can not be analyzed in this study. The analyzed AA should be total AA in the diet.

Author Response

Dear Reviewer, thank you for all comments to improve our manuscript. 

Please, find below our responses:

1. It should include the reason for choosing 20% higher Trp, Thr, and Met in the manuscript.

L510 - 513: Increasing dietary supplementation of Thr, Met and Trp in 20% related to Lys was done following Valini et al. [24] and Rodrigues et al. [25] trying to achieve similar beneficial results on performance and health such as found by them, however, using piglets under a mix management and poor sanitary conditions.

2. SID AA can not be analyzed in this study. The analyzed AA should be total AA in the diet.

It was solved on Table 1.

Reviewer 3 Report

Comments and Suggestions for Authors

.

Author Response

Please, consider this new version of the manuscript for publication.

thank you

Reviewer 4 Report

Comments and Suggestions for Authors

The authors made sufficient efforts to modify the manuscript.

Author Response

(The authors gave the same response as above.)
